# Exploring the Optimal Scale of Coastal Reclamation Activities Based on an Environmental Capacity Assessment System: A Case Study in Haizhou Bay, China

**Lan Feng [1,2,*], Xianyu Zeng [1], Pan Hu [1] and Xiaoxiao Xu [1]**

[1] College of Civil Engineering, Nanjing Forestry University, Nanjing 210037, China
[2] Ecological Complexity and Modeling Laboratory, Department of Botany and Plant Sciences, University of California, Riverside, CA 92521, USA
[*] Correspondence: fenglan0108@126.com

**Abstract:** With the acceleration of urbanization, the demand for land due to urban large-scale construction and development is increasing. Coastal reclamation (CR) is a prevailing approach to tackle the contradiction between coastal land shortage and the growing demand for living space for human beings. Enormous social and economic benefits are derived from CR, while at the same time bringing a series of environmental problems. Since the beginning of the 21st century, this oceanic-oriented development has become more frequent. Therefore, the considerable economic and ecological trade-offs of reclamation activities must be analyzed to enable targeted land use decisions. By comprehensively evaluating the natural conditions of the sea area, including geology, topography, hydrology, ecology, and social and economic conditions, this study established an environmental capacity assessment system (ECAS) based on water environmental capacity for assessment of the potential environmental impact resulting from CR. According to the water quality status and positions of CR in Haizhou Bay, the environmental capacities of four major pollutants were calculated to forecast the suitable area of CR. The results indicated that these reclamation projects had notable negative effects on the environmental capacity of the four major pollutants. The order of pollutants according to their harm on seawater quality is: $PO_4$-P > $NH_4$-N > $NO_X$-N > COD. In three reclamation alternative scenarios, scenario 3 led to the minimum negative impacts on the environmental capacity, scenario 2 followed, and scenario 1 had the worst result. Hence, scenario 3 was the optimal reclamation scenario, under which the suitable area of CR in Haizhou Bay was found to be 83 $km^2$. This study provides a scientific reference for the effective management of coastal reclamation and future environmental impact research when new CR is proposed, as well as sustainable urban development.

**Keywords:** coastal reclamation activities; comprehensive evaluation; sustainable urban development; environment capacity; Haizhou Bay





## 1. Introduction

As the point of interaction between sea and land, shoreline zones probably contain the most organically productive and diverse environments on our planet, such as coral reefs, estuaries and wetlands [1]. Seaside regions are significant and useful zones in light of their geographic location and plentiful regular assets that provide human existence and certain ventures [2,3]. The proportion of the marine economy in the national GDP expanded from 3.8% in 2003, to 9.0% in 2019, according to the Statistical Bulletin of the Chinese Marine Economy (SBCME) [4]. The sharp expansions of the salt, agriculture and fishery industries have produced enormous financial development for China [5]. Meanwhile, coastal zones have higher population densities, higher population growth rates and higher rates of urbanization [6]. In particular, China's continuous and rapid economic growth can cause land shortage problems. As important reserve land resources, tidal flats can significantly

alleviate land supply shortages and provide a large amount of land resources for coastal zones [7,8]. Subsequently, the exploitation and growth of coastal zones have enormously expanded as of late [6], and many tidal flats have been converted into salt pans, ports [9] and aquaculture ponds [10,11].

However, coastal environments are also economically and ecologically vulnerable areas threatened by various activities, such as sea level rise, pollution, reclamation and changes in storms associated with climate change [12,13]. Coastal reclamation (CR) projects have a profound history and occur widely in coastal areas, which has resulted in extensive changes in the location of the coastline along the entire coastline of China [14,15]. During this process, coastal environments and ecosystems have undergone serious intervention, and enormous coastal wetlands have been lost [14,16]. From 1950 to 2010, CR brought about a loss of 70% of coastal wetlands, with an average of 40,000 ha/a, increasing particularly dramatically after 2006 to provide for rapid urbanization development [17]. In addition, there has been loss of salt marshes and mangroves [18], and habitat degradation [19]. A large number of studies have begun to evaluate the ecological risks of CR in order to establish a restrictive mechanism for these activities. Murray et al. [20] detailed the extensive and enduring effect of rapid coastal development on international coastal environments, shadowing the delivery of significant marine ecosystem services. Zhang et al. [11] reported the dramatically expanding harm to natural habitat because of reclamation activities, in the past 20 years in Shanghai. Lin et al. [21] surveyed eight kinds of ecosystem service value and found that the total ecosystem service value diminished by approximately USD 5 billion over 20 years. Sengupta et al. [22] contrasted 16 worldwide megacities and the area of spread land, and found that seaside reclamation has decreased marine biodiversity and wetlands. Yang et al. [23] recognized that trade-offs existed among material creation, environment quality, and carbon storage, in a study of the Yellow River Delta from 1989 to 2015. As exemplified above, prior cases shared a concern of the ecological impact of CR, but did not give a clear answer on how to control the scale of CR. Peng et al. [24] proposed an estimation framework for reclamation area based on ecological and environmental cost, and analyzed the suitable area for CR. Qiu et al. [25] explored the progress of land reclamation in Hangzhou Bay over the past 30 years, analyzing the economic and ecological trade-offs of CR activities, with the hope of limiting reclamation activities. In 2012, the "Some Suggestions on Strengthening the Management of Sea for Regional Agricultural Reclamation" policy was announced [26], with strict guidelines adopted in the administration of ocean utilization for forestry, planting, animal husbandry and hydroponic projects. With these guidelines, the SOA planned to reinforce command over CR, ocean utilization and marine assurance in coastal zones. Consequently, the yearly reclamation activities showed a consistent downward pattern after 2013 [27]. However, whether the economic benefits of CR compensate for the resulting ecological losses has not yet been answered in any level of detail.

Therefore, there is an urgent need to exploit a novel approach to effectively deal with CR and to harmonize the contradiction between reclamation projects and coastal environment protection. One of the main solutions is to create a CR restriction mechanism, that is, a "red-line system", similar to the system in the nation's arable land protection policy [28]. The CR red line system involves establishing and codifying key areas suitable for reclamation, and allowable areas for reclamation in each particular region. Environmental costs associated with reclamation should be prioritized when estimating critical allowable reclamation areas. This paper attempts to establish an environmental capacity assessment system (ECAS) to calculate the optimum reclamation area for each specific sea zone. The main advantage of the ECAS approach described here, over other control management alternatives, lies in the fact that the development of an ECAS prioritizes the environment and takes into account the trade-offs between economic benefits and environmental costs. This paper hopes to provide basic information and scientific inspiration for the future policy of resource rolling and sustainable land use in coastal areas.

## 2. Materials and Methods

### 2.1. Study Area and Data

2.1.1. Study Area

Haizhou Bay is located on the eastern coast of Lianyungang City, Jiangsu Province (Figure 1), and adjacent to the southern coast of Rizhao City, Shandong Province. The bay mouth commences at Bergamot Tsui (35°05′55″ N, 119°21′53″ E) in Lanshan Town, Rizhao City in the north, and reaches Gaogong Island (34°45′25″ N, 119°29′45″ E) of Lianyungang City in the south, facing the Yellow Sea. It is 42 km wide, with a total coastline length of 86.81 km. The bay covers an area of 876.39 km$^2$, of which the area below 0 m is 687.90 km$^2$, the area below 5 m is 340.67 km$^2$, the area below 10 m is 63.01 km$^2$, and the maximum water depth is 12.20 m. The gulf coast is mainly silty silt coast, followed by bedrock coast, and sandy plain coast. The sea floor slopes gently from west to east.

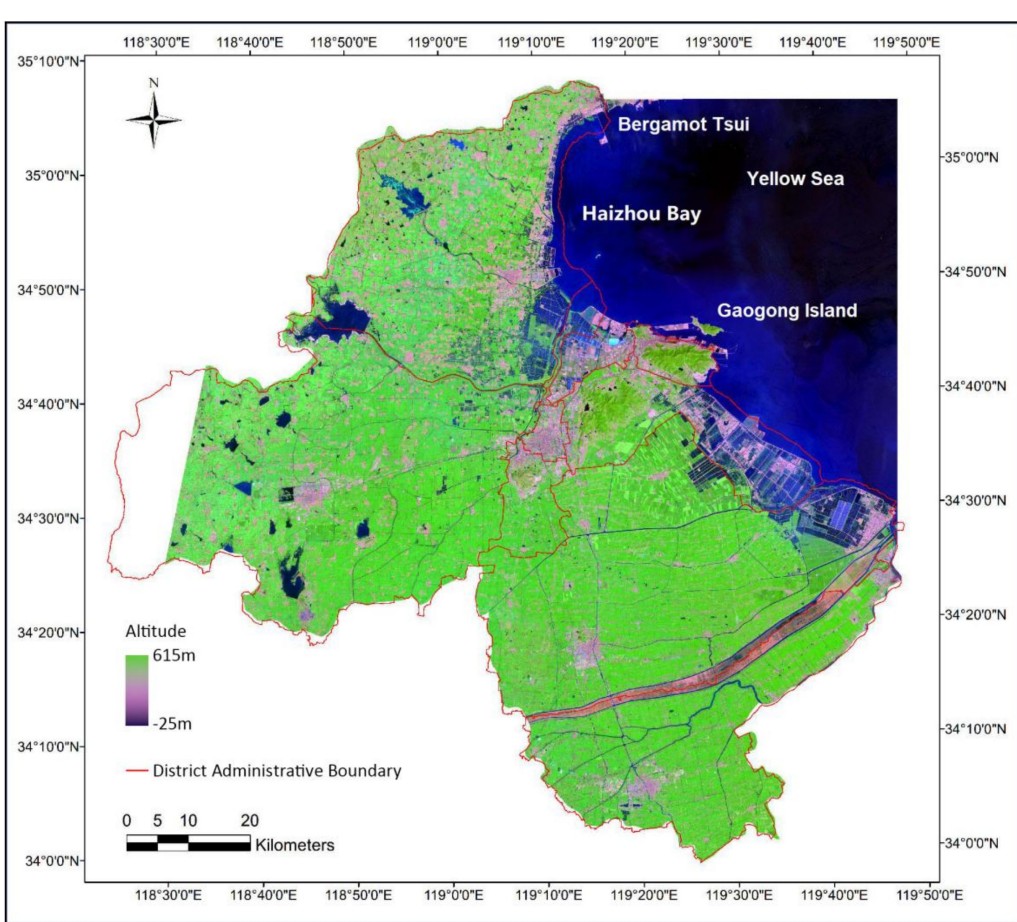

**Figure 1.** Map of the study area, Haizhou Bay, in 2016.

Haizhou Bay is bound by Laoyeding in the north, Yuntai Mountain in the south, and an alluvial marine plain in the west. The rivers entering the bay along the coast include Xiuzhen River, Longwang River, Qingkou River, Xinshu River and Qiangwei River; the latter two are called Linhong River after their convergence. Qinshan Island and the East–West Islands are located in Haizhou Bay, and Pingshan Island, Dashan Island and Cheniushan Island are located outside the bay. These islands are all bedrock islands. Transportation along the Gulf Coast is convenient: it is the eastern starting point of the Longhai railway, which crosses eastern China, and is the shortest sea exit for the nine provinces (autonomous regions) in the northwest and central plains. It is known as the "bridgehead at the eastern end of the Eurasian Continental Bridge", and is an important transportation hub linking north and south, and east and west, in China.

From 1985 to 2005, the area of tidal flat and wetland in Lianyungang City decreased by 25,000 hectares due to CR, accounting for 30% of the total coastal tidal flat area. In 2006, after Lianyungang City proposed the strategy of "advancing eastward and embracing the sea", the accumulated sea area of reclamation projects reached 473.8 km$^2$, accounting for 29% of the total tidal flat utilization area. From 2010 to 2012, a number of reclamation projects were implemented in the vicinity of Haizhou Bay, mainly in Ganyu County, Lianyun New Town, and other areas, adding 61.3 km$^2$ of land, including 31.2 km$^2$ of tidal flat and 17.67 km$^2$ of reclamation [29]. This behavior of indirectly changing wetland by CR caused damage of biological habitats and a decrease in the cultivation of commercial crops in wetlands, and had an extremely serious negative impact on fishery resources. For example, the number of natural fish species in Hailing Lake of Lianyungang decreased from 105 species in 1985, to 56 species in 2020.

Since 2010, water pollution in the coastal waters of Haizhou Bay has become increasingly serious, and seawater quality has generally been deteriorating. As a result of seawater pollution, the amount of biological resources declined and nutrient salt pollution became more serious. Inorganic nitrogen and phosphorus were found to be the main seawater pollutants, and highly polluted areas were the near-shore area and estuary [30]. The main manifestations of the impact of the reclamation projects on the sea area of Haizhou Bay were the change of the suspended sediment concentration in the water body, and the change of the self-purification capacity of the water body [31]. During the construction of the west levee, the semi-exchange period of the water body changed from one tidal period before the project, to 6.5 tidal periods after the project, which greatly reduced the hydrodynamic movement of the bay and weakened the water exchange capacity inside and outside the bay. The continuous accumulation of pollutants directly leads to the aggravation of water pollution in the bay [32]. Meanwhile, the construction of the west levee also led to the siltation of the beach, which directly threatened the tourism development of Haizhou Bay. In addition, during the process of reclamation, oil pollution from ships in some sea areas of the harbor was also serious, which damaged the coastal landscape to a certain extent, with adverse effects on biodiversity and the ecological environment of the bay [33].

Large-scale reclamation projects have directly changed the hydrological characteristics of the coastal waters and affected the migration rules of fish; indirectly, they have destroyed the habitat environment and spawning grounds of fish, destroyed the key ecological environment for many fish species' survival, and reduced fishing resources [31]. Particularly, near the bay, a reclamation project will directly affect the water exchange of the bay, and the salinity of the sea will also change when encountering a rainstorm, which will pose a great threat to the biological resources and aquaculture in the bay [34]. Consequently, the construction of the Lianyungang west levee has resulted in a reduction in the number of phytoplankton and zooplankton species in Lianyungang Harbor. In addition, coral reefs are a unique natural resource in Lianyungang. Years of CR activities have severely damaged part of the coral reefs, causing not only loss of their revetment function, tourism and other economic and social values, but have also caused the number of marine organisms that depend on coral reefs to reduce, or even disappear [35].

In accordance with the "Ganyu Port Area Master Plan" and "Jiangsu Marine Functional Zoning", this study selected 5 reclamation conditions from a total of 7 preparatory reclamation conditions (the two deleted reclamation conditions were located on an unsuitable shoreline, so were not considered), and the location and relevant description of each working condition are shown in Figure 2 and Table 1.

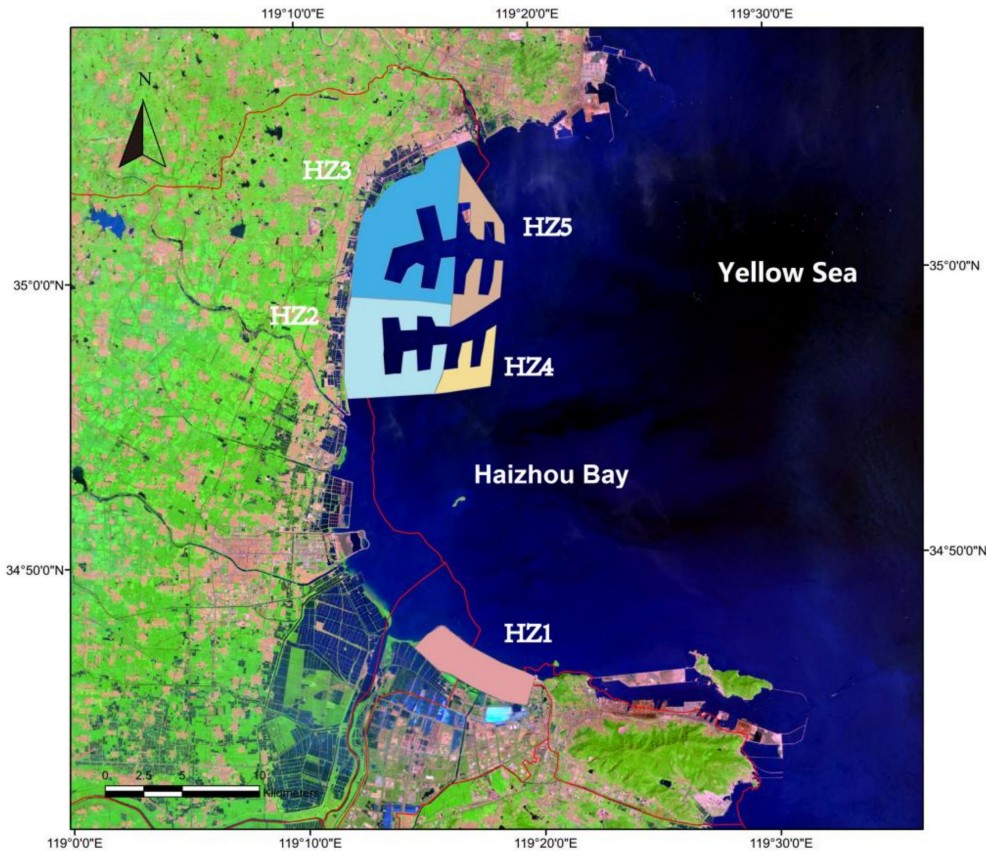

**Figure 2.** The geographic location of reclamation projects in Haizhou Bay (HZ1-HZ5 represent five projects).

**Table 1.** Description of reclamation projects.

| Working Condition | Area/km² | Usage | Reclamation Period |
|---|---|---|---|
| HZ1 | 18 | Urban development | 2010–2015 |
| HZ2 | 22 | Port construction | 2015–2030 |
| HZ3 | 46 | Port construction | 2015–2030 |
| HZ4 | 9 | Port construction | 2015–2030 |
| HZ5 | 19 | Port construction | 2015–2030 |

Data source: Lianyungang Ocean and Fishery Bureau and Lianyungang City Planning Bureau, 2012.

### 2.1.2. Study Data

The pollution sources entering the sea area of Haizhou Bay can be divided into two categories—sewage outfalls and rivers:

(1) Sewage outfalls are mainly responsible for sewage discharge from industrial enterprises along the coast of Haizhou Bay;

(2) Rivers are channels for clear water to enter the sea, and partly serve as sewage channels for land-based pollution into the sea.

### (1) Major Sewage Outfalls

The spatial distribution of the main sewage outfalls in Haizhou Bay is shown in Figure 3. The data of the main pollution sources and pollutant emissions in Haizhou Bay were drawn from "Lianyungang City Environmental Quality Report (2020)" (Table S1).

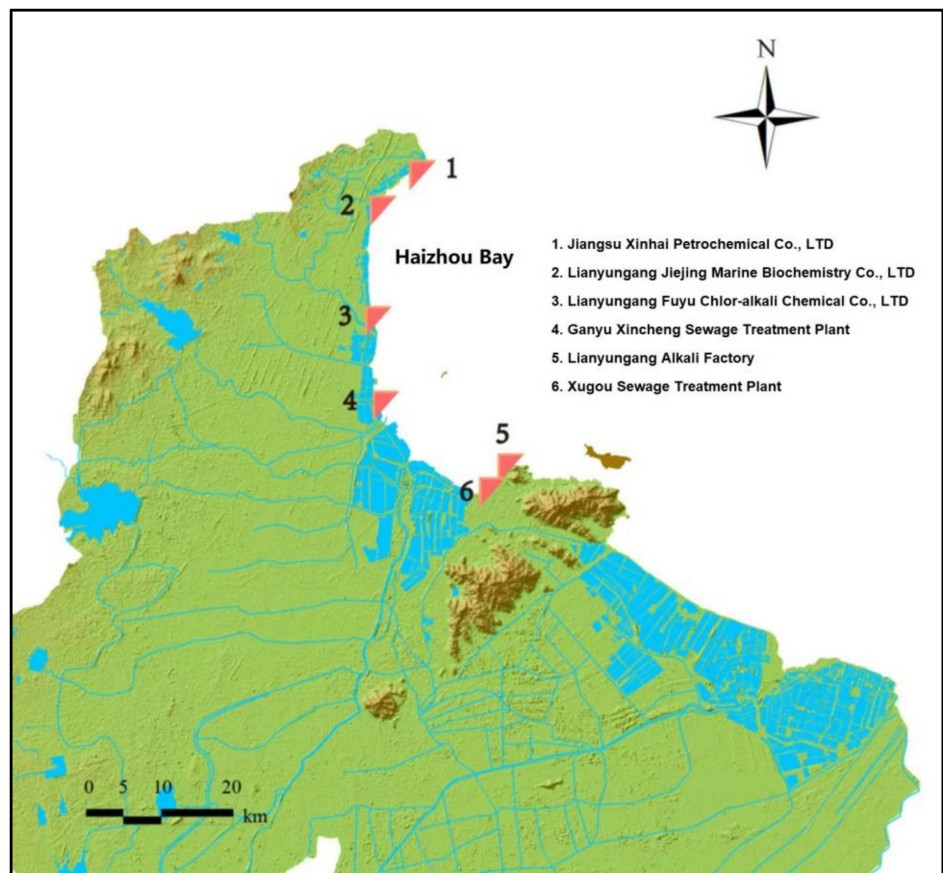

**Figure 3.** Distribution of main marine outfalls in Haizhou Bay.

In 2020, the largest sewage discharge source among industrial pollution sources was Lianyungang Soda Plant, with a discharge volume of $158.37 \times 104 \text{ m}^3/\text{a}$, and the largest discharge source of urban comprehensive sewage was Xugou Sewage Treatment Plant, with a discharge volume of $752.13 \times 104 \text{ m}^3/\text{a}$ (Table S1). In general, the total discharge of sewage treatment plants has decreased, and the discharge of various individual pollutants has also declined.

(2) Main Rivers

The spatial distribution of the estuary in Haizhou Bay is shown in in Figure 4. The data of the major pollutants into the sea in Haizhou Bay were drawn from "Lianyungang City Environmental Quality Report (2020)" (Table S2).

The flow of Linhong River still ranked first among all rivers entering the sea in Haizhou Bay in 2020, and its various pollutants entering the sea increased compared with those in 2019, except for total phosphorus (Table S2). In addition to the increase in the concentration of total nitrogen, the concentration of other pollutants in Longwang River decreased. In terms of the total amount of pollutants entering the sea, except for the increase in total nitrogen, the total amount of other pollutants all showed a downward trend. In this study, the statistical results of pollution sources in 2020 were used as the current pollution situation, to calculate the environmental capacity of each pollutant.

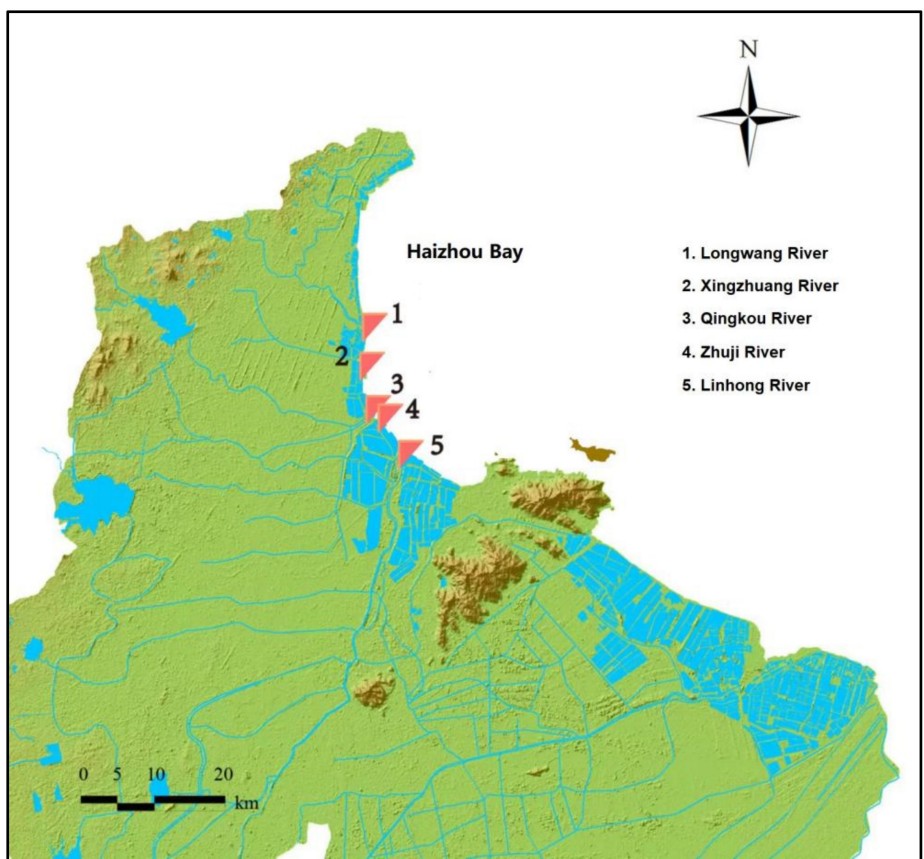

**Figure 4.** Distribution of main marine estuaries in Haizhou Bay.

*2.2. Environmental Capacity Assessment System (ECAS)*

2.2.1. Environmental Capacity Theory

Environmental capacity (EC) refers to a characteristic of the environment, which is the ability of the environment to accommodate a certain pollutant without causing an unbearable impact on the environment [36]. This feature contains three environmental implications: (1) pollutants existing in the environment will not affect the environment as long as they do not exceed the specified limit value; (2) no environment has an unlimited capacity to contain pollutants without affecting the physical and ecological functions of a specific ecosystem; and (3) EC can be quantified. Applying this concept to the ocean for further analysis, marine EC depends not only on the size of the space, location, tidal pattern, temperature and other hydrological conditions of the sea, as well as on physical, chemical, and biological migration and transformation conditions, but also depends on the environment quality standards that the sea should meet in order to maintain a certain sea environment function. In other words, the capacity of the marine environment is closely related to the natural background value of a particular sea area, the characteristics of various environmental elements, social functions, physical and chemical properties of pollutants, and the self-purification capacity of the marine environment, which can better reflect the impact of reclamation activities on the environment than water quality indicators. Therefore, we chose the EC of several typical pollutants in the coastal zone as the evaluation index, and conducted an in-depth consideration of the feasibility of the superimposed implementation of the reclamation scheme.

2.2.2. Environmental Capacity Assessment Indicators

In 2006, China began to implement a "total capacity control management" plan with reduction in chemical oxygen demand (COD) as the binding index. Therefore, COD has become the priority pollutant for total EC control in Haizhou Bay. In 2011, China added

ammonia nitrogen ($NH_4$-N) as a binding target to total capacity control, with an average annual emissions reduction of about 2%. In addition, according to the "China Marine Statistical Yearbook (2020)" and "Lianyungang Environmental Quality Bulletin (2020)", among the 18 water quality monitoring points in the coastal waters of Lianyungang, Class III sea water accounted for 16.67%, and Class IV sea water accounted for 33.33%. Among the ten seawater functional zones, five stated that their annual average value of indicators met the requirements of the corresponding functional zones, at a rate of 50%. The main pollutants were COD, inorganic nitrogen ($NO_x$-N) and active phosphate ($PO_4$-P). In Haizhou Bay, for the major rivers entering the sea, such as Longwang River, Xingzhuang River, Qingkou River and Linhong River, the main pollutants were COD and $NH_4$-N. Therefore, four pollutants, COD, $NH_4$-N, $NO_x$-N and $PO_4$-P, were selected for this study ($NO_x$-N refers to the sum of nitrate and nitrite content). The reduction in the EC of these four pollutants was used as the evaluation index of the reclamation schemes, that is, the reduction percentage of the EC of these pollutants relative to the current situation after the implementation of the reclamation scheme was calculated. The calculation formula is as follows:

$$Q = \frac{Q_b - Q_a}{Q_b} \times 100\% \tag{1}$$

where, $Q$ is the percentage reduction in EC for a pollutant in the bay; $Q_a$ is the EC of pollutants after reclamation(t/a); $Q_b$ is the EC of pollutants before reclamation(t/a); And $Q_C$, $Q_{NH3}$, $Q_N$, $Q_P$ represent the reduction percentage of EC of COD, $NH_4$-N, $NO_x$-N and $PO_4$-P, respectively.

2.2.2.1. Environmental Capacity Estimation

In this study, EC refers to the maximum allowable pollutant load in the bay without considering the allocation principle between pollution sources, only focusing on the attributes of the marine environment itself and the stipulated water quality objectives. The water quality of the grid points in the simulated area has a certain response relationship with the emission of each pollution source in a region, that is, the response relationship matrix between the pollutant concentration of monitoring points and the intensity of each emission source. The response relationship matrix is calculated by the following hydrodynamic model and water quality model. At the same time, the properties of the marine environment itself, such as hydrologic conditions and degradation capacity of the environment, are reflected in the response field of the pollution source when calculating the response relationship. Therefore, the calculation of EC can be transformed into a linear programming problem, with the water quality objective as a constraint condition and the pollutant discharge load as a target function [37]. The problem is to take the water quality target as the restriction condition, at the selected water quality control point, and calculate the maximum sum of the pollution load emissions of each sewage outlet; additionally, the pollutant concentration is prohibited from exceeding its corresponding environmental standard.

MIKE 21 is a professional two-dimensional free-surface flow water simulation software, which is applied to the planar simulation of hydraulic and related phenomena in estuaries, bays and coastal areas of the ocean. It can be used for the study of water flow, water environment change, and sediment transport in rivers, oceans and reservoirs [38]. The MIKE 21 two-dimensional tidal field model developed by the Danish Hydraulics Institute (DHI) was used to construct a two-dimensional hydrodynamic model of Haizhou Bay for simulating and calculating its water environment changes.

Hydrodynamic Model

1. Basic hydrodynamic model.

The model adopts the two-dimensional flow continuity equation and motion equation:

① Continuity equation:

$$\frac{\partial \zeta}{\partial t} + \frac{\partial p}{\partial x} + \frac{\partial q}{\partial y} = 0 \tag{2}$$

② Motion equations:

$$\frac{\partial p}{\partial t} + \frac{\partial}{\partial x}\left(\frac{p^2}{h}\right) + \frac{\partial}{\partial y}\left(\frac{pq}{h}\right) + gh\frac{\partial \zeta}{\partial x} + gp\frac{\sqrt{p^2 + q^2}}{C^2 h^2} - \Omega q - fVV_x = 0 \tag{3}$$

$$\frac{\partial q}{\partial t} + \frac{\partial}{\partial x}\left(\frac{q^2}{h}\right) + \frac{\partial}{\partial y}\left(\frac{pq}{h}\right) + gh\frac{\partial \zeta}{\partial y} + gq\frac{\sqrt{p^2 + q^2}}{C^2 h^2} - \Omega p - fVV_y = 0 \tag{4}$$

where:

$t$: time;

$g$: acceleration of gravity ($m/s^2$);

$\zeta(x, y, t)$: free water level (m);

$h(x, y, t)$: depth of the water (m), the distance from the bottom of the sea to the stationary sea surface;

$p, q(x, y, t)$: flow density in the $x-, y-$ direction, that is, the width of the flow ($m^3/s/m$);

$(u, v)$: $x-, y-$ directional perpendicular mean velocity component;

$C(x, y)$: the coefficient of Chezy, its relation to the Manning number M is $C = M \times h^{1/6}$;

$f(V)$: wind friction factor $= \gamma_\alpha^2 \rho_\alpha$; $\gamma_\alpha^2$ is the wind stress coefficient, $\rho_\alpha$ is the air density;

$V, V_x, V_y(x, y, t)$: wind speed, and $x-, y-$ component of wind speed in direction ($m/s$);

$\Omega(x, y)$: Coriolis coefficient $f = 2\omega \sin \phi$, $\omega$ is the rotational speed of the earth, $\phi$ is geographic latitude.

Equations (2)–(4) constitute the basic governing equations for solving the tidal flow field.

2. Boundary conditions.

In the numerical model adopted in this study, two kinds of boundary conditions should be given, namely, the open boundary condition and the closed boundary condition.

① Open boundary condition:

The open boundary condition is the water boundary condition. In this study, the tidal level is given as the open boundary. In order to obtain the change of the boundary water level with time, this model selected the actual monitoring data of the water level on the three endpoints of the boundary (Lanshan Port, Pingdao Island and Yanwei Port) as the basis data, and applied the model to interpolate the water levels of other points on the boundary line, to obtain the change data of the water level on the lines connecting Lanshan Port to Pingdao Island, and Pingdao Island to Yanwei Port. Due to the availability of data, the water level data of Lanshan Port and Yanwei Port were obtained from the "Tide Data 2019", and the water level of Pingdao Island was obtained from the tide conversion formula of the main port, Lianyungang, according to the water level data of Lianyungang in "Tide Data 2019".

② Closed boundary condition:

The closed boundary condition is the boundary condition between land and water. On this boundary, the normal flow velocity of the water quality point is 0. The time step of the model is finally determined by $\Delta t = 3600s$.

Water Quality Model

1. Basic water quality model.

The two-dimensional convection–diffusion transport model adopted in this model is as follows:

$$\frac{\partial}{\partial t}(hc) + \frac{\partial}{\partial x}(uhc) + \frac{\partial}{\partial x}(vhc) = \frac{\partial y}{\partial x}\left(h \cdot D_x \cdot \frac{\partial c}{\partial x}\right) + \frac{\partial y}{\partial x}\left(h \cdot D_y \cdot \frac{\partial c}{\partial y}\right) \tag{5}$$

where:

*h*: depth of the water (m);
*c*: concentration of pollutant (any unit);
*x, y*: horizontal velocity component in the x and y directions (m/s);
*Dx, Dy*: dispersion in the x and y directions (m$^2$/s);
*F*: linear attenuation coefficient (seconds $^{-1}$);
*S:Qs* ($c_s$-c);
*Qs*: source and sink;
*Cs*: pollutant emission concentrations in sources and sinks. The flow velocity and other information (including *u, v* and *h*) at each time step are provided by the hydrodynamic module.

2.    The boundary conditions.

① Open boundary conditions:
The open boundary condition refers to the concentration of pollutants in seawater exchanged with seawater outside the open boundary. When calculating the water exchange rate of the reclamation schemes, COD was used as the tracer pollutant, and the initial value of COD concentration inside Haizhou Bay was set as 1 mg/L, and outside Haizhou Bay as 0 mg/L.

② Closed boundary condition:
The closed boundary condition is the boundary condition between land and water. Zero flux boundary conditions were used for this boundary.

3.    Related parameters.

① Degradation coefficient: The coefficient is related to the nature of a particular pollutant and can be adjusted during model calibration. When calculating the water exchange rate of the reclamation scheme, COD was used as the tracer pollutant and the degradation coefficient was set as 0.

② Dispersion coefficient: Dispersion coefficient is the rate coefficient that represents the dispersion of pollutants in the flowing water along the direction of water flow, and the unit is square meters per second. When calculating the water exchange rate of the reclamation schemes, the calculation rate of dispersion coefficient was set as 18.7 m$^2$/s.

Linear Programming Model

The linear programming model is expressed as:

$$\max Z = Q^T X \tag{6}$$

required to meet:

$$\begin{cases} AX + B \leq S \\ X_i \leq X \leq X_l \\ \quad X \geq 0 \end{cases} \tag{7}$$

$$A = \begin{bmatrix} a_{11} & \cdots & a_{1n} \\ \cdots & a_{ij} & \cdots \\ a_{m1} & \cdots & a_{mn} \end{bmatrix} \tag{8}$$

where:

*Z*: objective function;
*Q*: coefficient vector; when the total amount of pollutants reaches the maximum value, $Q = [1, 1, \ldots , 1]^T$;
*A*: response coefficient matrix, generated by the concentration response field of pollution sources and the water quality monitoring points;
$a_{ij}$: pollution contribution coefficient of unit load of the $j_{th}$ sewage outlet to the $i_{th}$ water quality monitoring point;
*m*: number of water quality monitoring points;

*n*: number of pollution sources;

*B*: pollutant background concentration vector, $B = [b_1, b_2, \ldots, b_m]^T$;

*S*: standard vector of water quality, $S = [s_1, s_2, \ldots, s_m]^T$;

*X*: pollutants discharge load, $X = [x_1, x_2, \ldots, x_n]^T$; $X_i$: the lower limit vector of pollutant discharge load, $X_i = [x_{i1}, x_{i2}, \ldots, x_{in}]^T$; $X_j$: the upper limit vector of pollutant discharge load, $X_j = [x_{j1}, x_{j2}, \ldots, x_{jn}]^T$;

Equation (6): objective function equation;

Equation (7): constraint equations of water quality and pollution source load.

The verification results of the hydrodynamic model and water quality model in this paper are shown in the Supplementary Materials.

Determination of Water Quality Objectives

According to the classification system and classification standards of "Technical Guidelines for Marine Function Zoning", "National Marine Function Zoning", "Jiangsu Province Marine Function Zoning (2011–2020)" and "Lianyungang City Marine Function Zoning (2013–2020)", the marine function zoning of Lianyungang adopted a five-category and four-level system which was divided into five categories: development and utilization areas, regulation and utilization areas, marine protection areas, special function areas, and reserved area.

In this study, 13 routine water quality monitoring points of the Environmental Monitoring Center were selected, and their water quality targets were determined according to their zoning locations (Table 2). In Haizhou Bay, 12 main discharge outlets of pollution sources and estuaries of rivers entering the sea were selected to calculate the response coefficient matrix of pollution sources to water quality monitoring points (Figure 5).

Response Coefficient Field

Due to the transport and diffusion characteristics of ocean water, the distribution of response coefficient values in the sea will vary with different locations, thus, forming a response coefficient field. It reflects the response relationship of water quality to a pollution source. In view of 12 important pollution sources in Haizhou Bay, the response coefficient fields of each pollution source to water quality monitoring points after the implementation of each reclamation project were calculated, and the influence of each reclamation project on the distribution of pollutant response concentration fields was analyzed. Figure 6 shows the response concentration fields of COD formed by 12 pollution sources under different reclamation schemes in scenario 3.

**Table 2.** Water quality target at monitoring points.

| Water Quality Monitoring Point | Function Description | Water Quality Standard |
|:---:|:---:|:---:|
| 1 | Storm surge area | Class II |
| 2 | Reserved tourist area | Class III |
| 3 | Marine fishing area | Class II |
| 4 | No fishing area | Class II |
| 5 | Sewage discharge area | Class IV |
| 6 | Mariculture area | Class II |
| 7 | Mariculture area | Class II |
| 8 | Haizhou Bay Tourist Resort | Class III |
| 9 | Lianyungang Port area | Class IV |
| 10 | Coastal salt farming area | Class II |
| 11 | Lianyungang Port area | Class IV |
| 12 | Fishery breeding area | Class II |
| 13 | Storm surge area | Class IV |

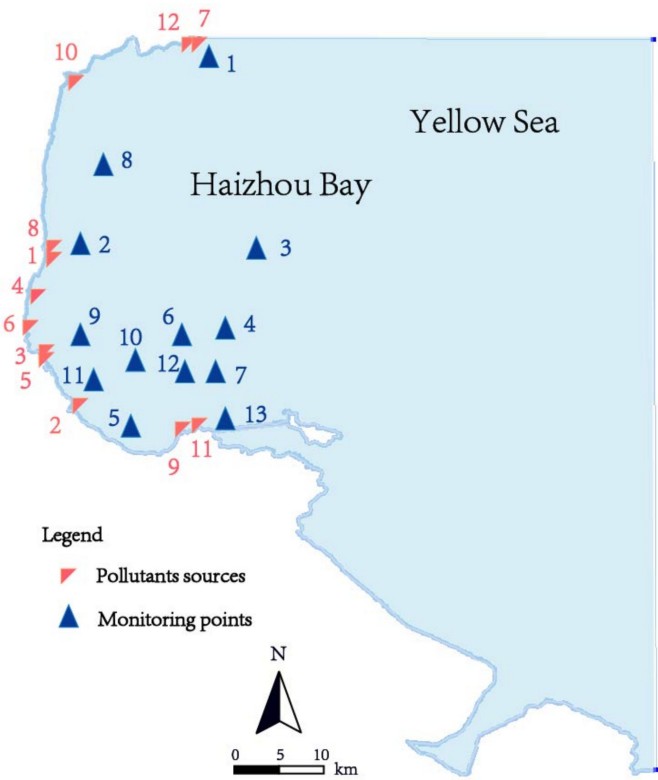

**Figure 5.** Pollution sources and water quality monitoring sites (sources: 1–5, river estuaries; 6, wastewater treatment plant outfalls; 7, petrochemical company; 8, chemical company; 9, alkali factory; 10, biochemical company; 11, wastewater treatment plant outfalls; 12, wastewater treatment plant outfalls).

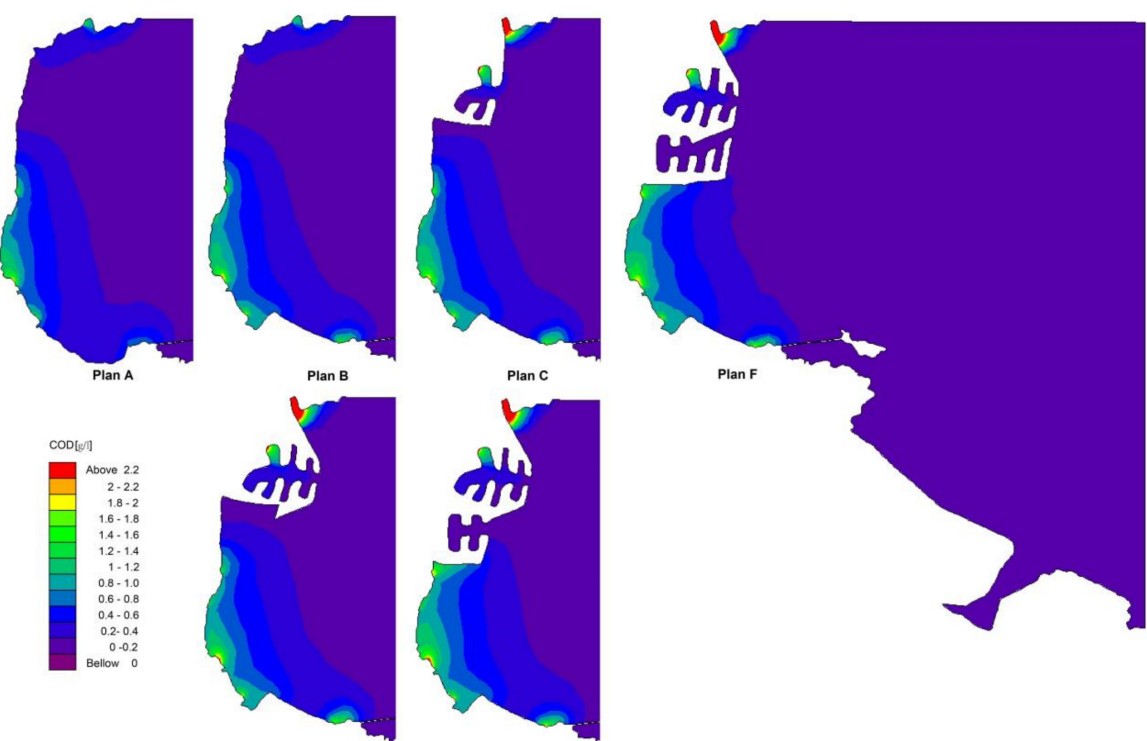

**Figure 6.** The COD response fields for 12 pollution sources (scenario 3). The COD response field in plan A; the COD response field in plan B; the COD response field in plan C; the COD response field in plan D; the COD response field in plan E; the COD response field in plan F.

### 2.2.3. Comprehensive Assessment

In the formulation and implementation of a TMDL plan, the U.S. Environmental Protection Agency has established corresponding provisions: when water quality exceeds 10% of the water quality index value stipulated by the Clean Water Act, the water body will be threatened by pollutants and the water quality will be considered as unsafe [39]. In addition, during the calculation of water EC, in order to cope with the uncertain risks existing in the future utilization of water resources and ensure the attainment of environmental water quality objectives, a certain proportion, namely, the safety margin of water EC, should be reserved in advance in the allocation of EC [40]. In the setting of the safety margin, the range of safety margin is related to the degradation speed of the harmful or toxic degradable pollutants; the higher the degradation speed, the safer it is. For general degradable pollutants, the degradation rate can be reduced by 10% or 20% (the stricter rate is selected from existing data experience), or the standard value of water quality target can be reduced by 10%, or the allocation of key pollution sources can be reduced by 10% [41]. In this study, the evaluation indexes of EC were all non-toxic or non-harmful pollution factors, and for Haizhou Bay, there were no empirical data as the reference standard. Therefore, the variation of EC of these four pollutants was set at 10% of their EC. Ultimately, 10% was set as the upper limit of the change in EC, that is, reclamation would be strictly prohibited if its impact on EC exceeds 10%.

In view of the specific pollutant discharge method and environmental quality of Haizhou Bay, we introduced an environmental capacity feasibility assessment index, and established an environmental capacity assessment system (ECAS) to quantitatively evaluate the environmental impact of all possible reclamation schemes. The evaluation system divided the feasibility of the schemes into three grades: feasible, basically feasible and infeasible (Table 3). $R$ is the comprehensive evaluation index, which was determined by the average value of all indicators. When $R \geq 8$, the reclamation scheme was feasible. When $8 > R \geq 4$, the reclamation scheme was basically feasible. When $R < 4$, reclamation was not feasible. Finally, feasible and basically feasible schemes were the acceptable reclamation schemes in this study, which were included in the calculation of the appropriate scale reclamation schemes.

**Table 3.** Feasibility assessment system based on marine EC.

| Index | Standard | Evaluation Score | Feasibility |
|---|---|---|---|
| $Q_C$ (t/a) | $\leq 4$ | 10 | |
| | $\leq 8$ | 8 | |
| | $\leq 10$ | 4 | |
| $Q_{NH3}$ (t/a) | $\leq 4$ | 10 | $R = \text{average}\{R_1, R_2, R_3, R_4\}$ |
| | $\leq 8$ | 8 | $R \geq 8$, feasible |
| | $\leq 10$ | 4 | $8 > R \geq 4$, basically feasible |
| $Q_N$ (t/a) | $\leq 4$ | 10 | $R < 4$, infeasible |
| | $\leq 8$ | 8 | |
| | $\leq 10$ | 4 | |
| $Q_P$ (t/a) | $\leq 4$ | 10 | |
| | $\leq 8$ | 8 | |
| | $\leq 10$ | 4 | |

## 3. Results and Discussion

### 3.1. Scenario 1 (S1)

The scenario is the superposition of different working conditions. Scenario 1 is the superposition of working conditions in the following order: superimposing from HZ1, HZ2, HZ3, HZ4 to HZ5 (Table 4).

**Table 4.** Changes in the sea area of Haizhou Bay (scenario 1).

| Reclamation Scheme | Working Conditions | Reclamation Area/km$^2$ | Calculation Area of EC/km$^2$ |
|---|---|---|---|
| A | Current situation (no working condition) | 0.00 | 876.39 |
| B | HZ1 | 18 | 858.39 |
| C | HZ1+HZ2 | 40 | 836.39 |
| D | HZ1+HZ2+HZ3 | 86 | 790.39 |
| E | HZ1+HZ2+HZ3+HZ4 | 95 | 781.39 |
| F | HZ1+HZ2+HZ3+HZ4+HZ5 | 114 | 762.39 |

### 3.1.1. Calculation Area of EC

The scope of the sea area of Haizhou Bay for calculating the current value of EC in this study comprised from the Bergamot Tsui (35°05′55″ N, 119°21′53″ E) in Lanshan Town, Rizhao City, Shandong Province, in the north, to Gaogong Island (34°45′25″ N, 119°29′45″ E) in Lianyungang District, Lianyungang City, Jiangsu Province, in the south, to the connecting line of the two places in the east, and to the coastline of Haizhou Bay in the west (Figure 7 plan A). The EC of pollutants in Haizhou Bay were calculated according to the coastal shape formed by the superposition of reclamation working conditions (Figure 7 plans B–F).

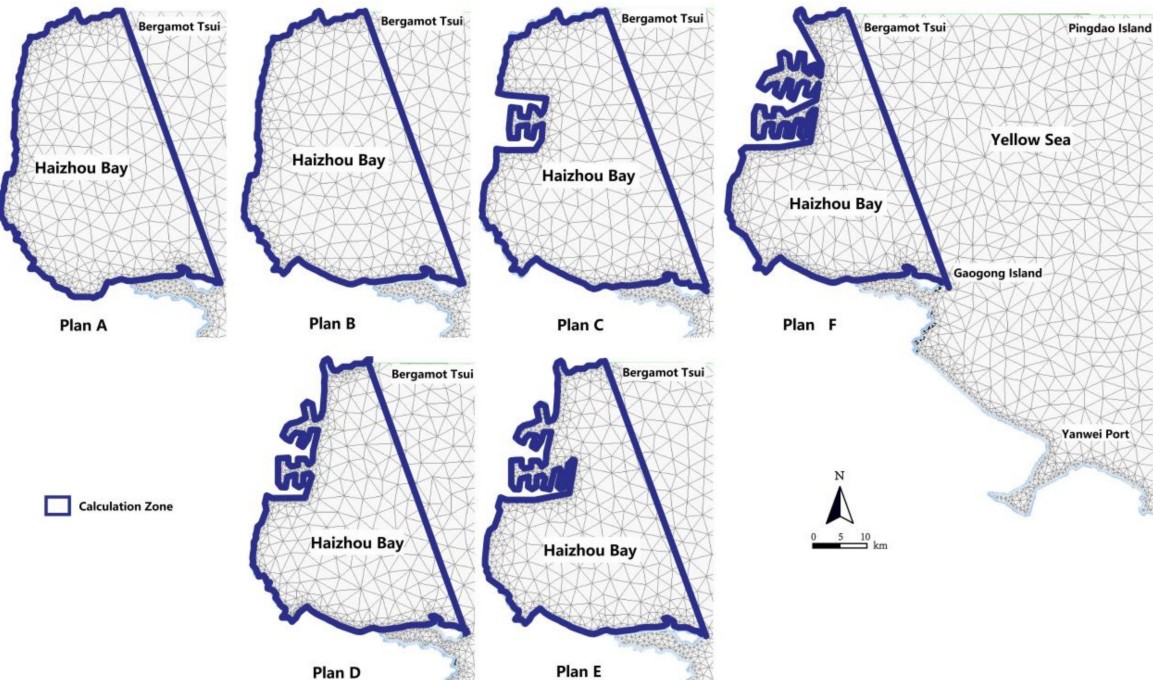

**Figure 7.** Computed region of marine EC (scenario 1). Computed region of plan A; computed region of plan B; computed region of plan C; computed region of plan D; computed region of plan E; computed region of plan F.

Working condition HZ1 was used for urban development, so the project area was not very large and the shape of the reclamation was relatively regular. Working conditions HZ2-HZ5 were used for port construction, so the shapes formed by reclamation were irregular polygons. The reclamation project will result in changes in the sea area of Haizhou Bay (Table 4).

### 3.1.2. COD

In the calculation of COD environmental capacity, after the implementation of scheme B (HZ1), the COD environmental capacity reduced by 1.48% (Table 4). This indicated that reclamation condition HZ1, which covered an area of 18 km$^2$, had a small impact

on the diffusion movement of pollutants, so it had little impact on the change of EC. The implementation of scheme C (HZ1+HZ2) caused the change of COD environmental capacity to reach 21.19%. The position of working condition HZ2 was located almost in the middle of the shoreline of Haizhou Bay, which was equivalent to separating Haizhou Bay across the middle after implementation, and may directly affect the diffusion of pollutants in the northern and southern parts of Haizhou Bay, thus, leading to large changes in EC. Meanwhile, implementation of schemes D, E and F had a huge impact on COD environmental capacity, especially D and E, each of whose impact exceeded 20% (Table 5). This showed that the effect of superposition implementation of working conditions on EC increased gradually. However, scheme F reduced the EC to some extent, probably because after the implementation of all working conditions, the concentration of pollutants in the north decreased, while the diffusion movement of pollutants in the south accelerated, leading to a moderate increase in EC.

**Table 5.** Feasibility assessment in COD environmental capacity.

| Reclamation Scheme | COD (t/a) | EC Change (%) | $R_1$ |
|---|---|---|---|
| A | 62,444 | - | - |
| B | 61,517 | 1.48 | 10 |
| C | 49,213 | 21.19 | 0 |
| D | 47,803 | 23.44 | 0 |
| E | 49,657 | 20.48 | 0 |
| F | 52,314 | 16.22 | 0 |

The feasibility evaluation results of schemes B to F are shown in Table 4. The results showed that the evaluation index $R_1$ of scheme B was 10, so scheme B was feasible. The evaluation index $R_1$ of schemes C, D, E and F was 0, so schemes C, D, E and F were not feasible, that is, working conditions HZ2, HZ3, HZ4 and HZ5 were prohibited.

### 3.1.3. $NH_4$-N

In the calculation of $NH_4$-N environmental capacity, after the implementation of scheme B (HZ1), the EC reduced by 4.45% (Table 6). The influence was larger than the change in COD environmental capacity (1.48%), indicating that reclamation condition HZ1 had a slightly greater influence on the diffusion movement of $NH_4$-N pollutants. The implementation of scheme C (HZ1+HZ2) caused the change of $NH_4$-N environmental capacity to reach 21.43%. This indicated that the position of HZ2 as a working condition also had a large influence on the diffusion of $NH_4$-N pollutants. The implementation of schemes D, E and F had the same effect on the EC of $NH_4$-N as the change of COD environmental capacity.

**Table 6.** Feasibility evaluation in $NH_4$-N environmental capacity.

| Reclamation Scheme | $NH_4$-N(t/a) | EC Change (%) | $R_2$ |
|---|---|---|---|
| A | 3366 | - | - |
| B | 3217 | 4.45 | 8 |
| C | 2645 | 21.43 | 0 |
| D | 2564 | 23.83 | 0 |
| E | 2664 | 20.87 | 0 |
| F | 2831 | 15.90 | 0 |

The feasibility evaluation results of $NH_4$-N environmental capacity are shown in Table 5. The results demonstrated that the evaluation index $R_2$ of scheme B was 8, so scheme B was basically feasible. The evaluation index $R_2$ of schemes C, D, E and F was 0, so schemes C, D, E and F were not feasible, that is, the working conditions HZ2, HZ3, HZ4 and HZ5 were prohibited.

### 3.1.4. $NO_X$-N

In the calculation of $NO_X$-N environmental capacity, after the implementation of scheme B (HZ1), the EC reduced by 3.84% (Table 7). This impact was not much different from the change of COD environmental capacity (1.48%), and the impact was small. The implementation of schemes C, D, E and F had a great impact on the change of $NO_X$-N environmental capacity, all of which exceeded 18%. Different from COD and $NH_4$-N environmental capacity, the first two pollutants were the most sensitive to scheme D, and the EC changed the most, while $NO_X$-N was most sensitive to scheme E, with a change of 25.38%.

**Table 7.** Feasibility evaluation in $NO_X$-N environmental capacity.

| Reclamation Scheme | $NO_X$-N(t/a) | EC Change (%) | $R_3$ |
|---|---|---|---|
| A | 5082 | - | - |
| B | 4887 | 3.84 | 10 |
| C | 4139 | 18.56 | 0 |
| D | 3855 | 24.12 | 0 |
| E | 3792 | 25.38 | 0 |
| F | 4021 | 20.88 | 0 |

The feasibility assessment results of $NO_X$-N environmental capacity are shown in Table 6. The results indicated that the evaluation index $R_3$ of scheme B was 10, so scheme B was feasible; the evaluation index $R_3$ of schemes C, D, E and F was 0, so schemes C, D, E and F were not feasible, that is, working conditions HZ2, HZ3, HZ4 and HZ5 were prohibited.

### 3.1.5. $PO_4$-P

The impact of the reclamation scheme on PO4-P environmental capacity was similar to those of the previous three pollutants. Scheme B (HZ1) had the least impact on the change of EC, of only 2.58% (Table 8); However, for schemes C, D, E and F, $PO_4$-P environmental capacity changes were relatively huge, all exceeding 10%. At the same time, scheme D had the greatest impact on EC, which was 23.87%.

**Table 8.** Feasibility evaluation in $PO_4$-P environmental capacity.

| Reclamation Scheme | $PO_4$-P(t/a) | EC Change (%) | $R_4$ |
|---|---|---|---|
| A | 421 | - | - |
| B | 410 | 2.58 | 10 |
| C | 373 | 11.49 | 0 |
| D | 321 | 23.87 | 0 |
| E | 345 | 18.16 | 0 |
| F | 332 | 21.11 | 0 |

The feasibility evaluation results of $PO_4$-P environmental capacity demonstrated that the evaluation index $R_4$ of scheme B was 10, so scheme B was feasible; The evaluation index $R_4$ of schemes C, D, E and F were all 0, so schemes C, D, E and F were not feasible, and working conditions HZ2, HZ3, HZ4 and HZ5 were prohibited.

### 3.1.6. Comprehensive Evaluation of Feasibility

According to the analysis of the above four indicators, in scenario 1, the reclamation scheme had a large impact on the EC of pollutants. Only scheme B was awarded the evaluation score, and the comprehensive evaluation index R was 9.5, while the other schemes were all 0 (Table 9). Therefore, only working condition HZ1 was feasible, and working conditions HZ2, HZ3, HZ4 and HZ5 were strictly restricted.

**Table 9.** Feasibility evaluation on EC (scenario 1).

| Reclamation Scheme | $R_1$ | $R_2$ | $R_3$ | $R_4$ | $R$ | Evaluation Result |
|---|---|---|---|---|---|---|
| A | - | - | - | - | - | - |
| B | 10 | 8 | 10 | 10 | 9.5 | feasible |
| C | 0 | 0 | 0 | 0 | 0 | infeasible |
| D | 0 | 0 | 0 | 0 | 0 | infeasible |
| E | 0 | 0 | 0 | 0 | 0 | infeasible |
| F | 0 | 0 | 0 | 0 | 0 | infeasible |

According to the results of the comprehensive feasibility evaluation, in this evaluation, scheme B was acceptable, and schemes C, D, E and F were not feasible. Therefore, the suitable scale of reclamation in Haizhou Bay was 18 km$^2$ (Table 10).

**Table 10.** The maximum allowable area for CR in Haizhou Bay (scenario 1).

| Reclamation Scheme | Evaluation Result | Area (km$^2$) | Appropriate Scale of Reclamation (km$^2$) |
|---|---|---|---|
| A | - | 0.00 | |
| B | feasible | 18 | |
| C | infeasible | 40 | |
| D | infeasible | 86 | 18 |
| E | infeasible | 95 | |
| F | infeasible | 114 | |

*3.2. Scenario 2 (S2)*

The difference between scenario 2 and scenario 1 was that in the order of superimposed working conditions, HZ1 and HZ3 were superimposed first, then HZ2 and HZ5, superimposing from HZ1, HZ3, HZ2, HZ5 to HZ4 (Table 11).

**Table 11.** Changes in marine area of Haizhou Bay (scenario 2).

| Reclamation Scheme | Working Conditions | Reclamation Area/km$^2$ | Calculated Area of EC/km$^2$ |
|---|---|---|---|
| A | Current situation (no working condition) | 0.00 | 876.39 |
| B | HZ1 | 18 | 858.39 |
| C | HZ1+HZ3 | 64 | 812.39 |
| D | HZ1+HZ2+HZ3 | 86 | 790.39 |
| E | HZ1+HZ2+HZ3+HZ5 | 105 | 771.39 |
| F | HZ1+HZ2+HZ3+HZ4+HZ5 | 114 | 762.39 |

3.2.1. Calculation area of EC

The current value of scenario 2 was the same as scenario 1 (Figure 8). The EC of pollutants in Haizhou Bay in scenario 2 was calculated according to the coastal shape formed by the superposition of the working conditions (Figure 8).

In scenario 2, various reclamation projects will also cause changes in the sea area of Haizhou Bay (Table 11).

3.2.2. COD

In Table 12, scheme B was the same as in scenario 1. After implementation, COD environmental capacity reduced by 1.48%. There was a large difference between scheme C and scenario 1. After the implementation of scheme C, COD environmental capacity only reduced by 0.15%, indicating that the location of reclamation conditions had a huge impact on the EC. Schemes D, E and F were the same as scenario 1, which had a large impact on the EC, and COD environmental capacity was reduced by more than 16%.

The feasibility evaluation results of EC indicated that the evaluation index $R_1$ of schemes B and C was 10, so schemes B and C were feasible (Table 12). The evaluation index $R_1$ of schemes D, E and F was 0, so schemes D, E and F were not feasible, that is, working conditions HZ2, HZ4 and HZ5 were prohibited.

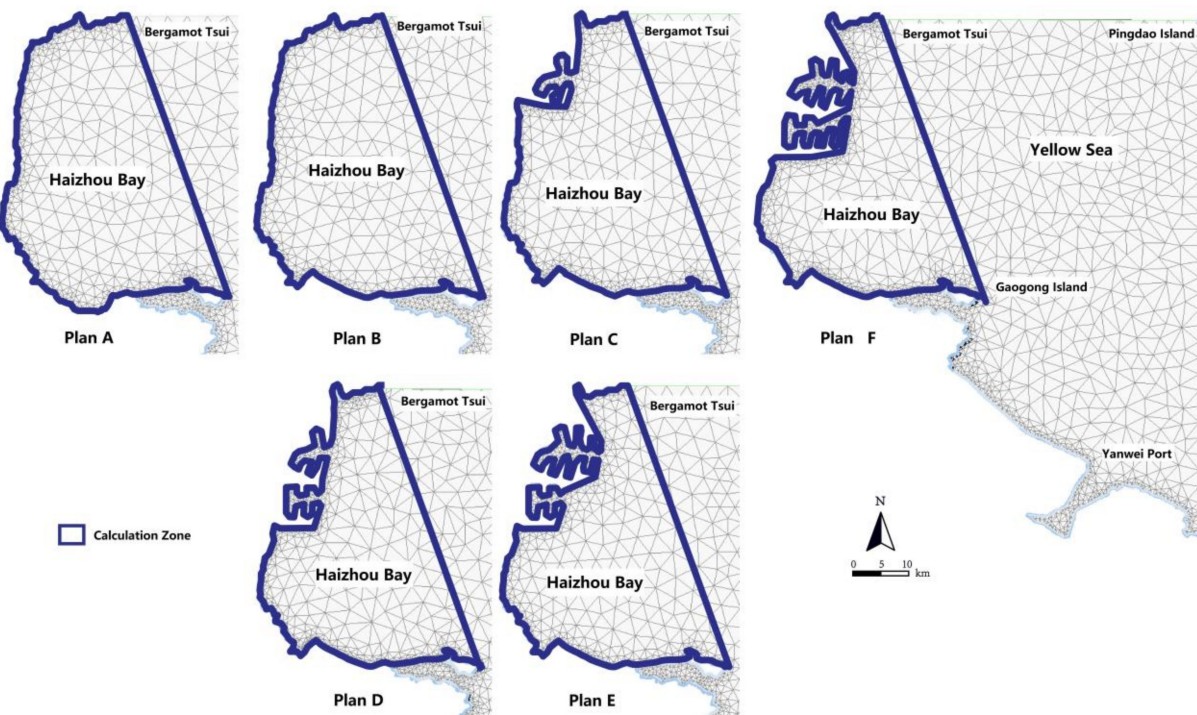

**Figure 8.** Computed region of marine EC (scenario 2). Computed region of plan A; computed region of plan B; computed region of plan C; computed region of plan D; computed region of plan E; computed region of plan F.

**Table 12.** Feasibility evaluation in COD environmental capacity.

| Reclamation Scheme | COD (t/a) | EC Change (%) | $R_1$ |
|---|---|---|---|
| A | 62,444 | - | - |
| B | 61,517 | 1.48 | 10 |
| C | 62,350 | 0.15 | 10 |
| D | 47,803 | 23.44 | 0 |
| E | 48,584 | 22.20 | 0 |
| F | 52,314 | 16.22 | 0 |

### 3.2.3. NH$_4$-N

Scenario 2 was the same as scenario 1. After the implementation of scheme B (HZ1), the EC reduced by 4.45% (Table 13). The implementation of scheme C, which was different from scenario 1, changed the impact on EC due to different working conditions. Scheme C had little impact on the change of NH$_4$-N environmental capacity, only 0.62%, which was not very different from the change of COD environmental capacity (0.15%). In schemes D, E and F, the EC of NH$_4$-N was reduced by 23.83%, 20.59% and 15.90%, respectively, indicating that the implementation of these schemes had an extensive impact on the surrounding environment.

The feasibility evaluation results of NH$_4$-N environmental capacity indicated that the evaluation index $R_2$ of scheme B was 8, so scheme B was basically feasible; the evaluation index $R_2$ of scheme C was 10, therefore, scheme C was also feasible. The evaluation index $R_2$ of schemes D, E and F was 0, and, therefore, schemes D, E and F were not feasible, that is, working conditions HZ2, HZ4 and HZ5 were prohibited.

**Table 13.** Feasibility evaluation in $NH_4$-N environmental capacity.

| Reclamation Scheme | $NH_4$-N (t/a) | EC Change (%) | $R_2$ |
|---|---|---|---|
| A | 3366 | - | - |
| B | 3217 | 4.45 | 8 |
| C | 3346 | 0.62 | 10 |
| D | 2564 | 23.83 | 0 |
| E | 2673 | 20.59 | 0 |
| F | 2831 | 15.90 | 0 |

### 3.2.4. $NO_X$-N

In scenario 2, scheme B was the same as scenario 1, and the $NO_X$-N environmental capacity reduced by 3.84% (Table 14). As for the first two pollutants, scheme C also had little impact on the $NO_X$-N environmental capacity, which only reduced by 0.07%, indicating that the implementation of working condition HZ3 had a weak impact on the surrounding environment. Schemes D, E and F, similar to scenario 1, reduced the EC by more than 16%. However, in scheme E, compared with scenario 1, the reduction in EC increased slightly, which may have been due to the accelerated diffusion of pollutants in the surrounding sea areas after the implementation of working condition HZ4.

**Table 14.** Feasibility evaluation in $NO_X$-N environmental capacity.

| Reclamation Scheme | $NO_X$-N (t/a) | EC Change (%) | $R_3$ |
|---|---|---|---|
| A | 5082 | - | - |
| B | 4887 | 3.84 | 10 |
| C | 5045 | 0.07 | 10 |
| D | 3855 | 24.12 | 0 |
| E | 4250 | 16.37 | 0 |
| F | 4021 | 20.88 | 0 |

The results demonstrated that the evaluation index $R_3$ of schemes B and C was 10, so schemes B and C were feasible; the evaluation index $R_3$ of schemes D, E and F was 0, so schemes D, E and F were not feasible, and working conditions HZ2, HZ4 and HZ5 were prohibited.

### 3.2.5. $PO_4$-P

In scenario 2, the reclamation conditions of scheme B were the same as scenario 1, so the impact on $PO_4$-P environmental capacity was consistent with scenario 1, at only 2.58% (Table 15). Scheme C, different from the first three pollutants, had a wide impact on $PO_4$-P environmental capacity, reaching 5.19%. Schemes D, E and F, similar to scenario 1, had a significant reduction in EC, reaching more than 15%. However, compared with scenario 1, the impact of scheme E on $PO_4$-P environmental capacity slightly reduced, by 2.28%.

**Table 15.** Feasibility evaluation in $PO_4$-P environmental capacity.

| Reclamation Scheme | $PO_4$-P(t/a) | EC Change (%) | $R_4$ |
|---|---|---|---|
| A | 421 | - | - |
| B | 410 | 2.58 | 10 |
| C | 399 | 5.19 | 8 |
| D | 321 | 23.87 | 0 |
| E | 354 | 15.88 | 0 |
| F | 332 | 21.11 | 0 |

The EC feasibility evaluation results demonstrated that the evaluation index $R_4$ of scheme B was 10, and scheme B was feasible; the evaluation index $R_4$ of scheme C was 8,

and scheme C was also basically feasible. Similar to the first three pollutants, schemes D, E and F were not feasible, and working conditions HZ2, HZ4 and HZ5 were prohibited.

3.2.6. Comprehensive Evaluation of Feasibility

In scenario 2, according to the analysis of the above four indicators, reclamation schemes D, E and F all had a giant impact on the EC of pollutants (Table 16), so the comprehensive evaluation index was 0. Schemes B and C had little impact on the EC of these four pollutants, and the comprehensive evaluation index $R$ was 9.5. Therefore, working conditions HZ1 and HZ3 were feasible, while HZ2, HZ4 and HZ5 were all forbidden.

**Table 16.** Feasibility evaluation on EC (scenario 2).

| Reclamation Scheme | $R_1$ | $R_2$ | $R_3$ | $R_4$ | $R$ | Evaluation Result |
|:---:|:---:|:---:|:---:|:---:|:---:|:---:|
| A | - | - | - | - | - | - |
| B | 10 | 8 | 10 | 10 | 9.5 | feasible |
| C | 10 | 10 | 10 | 8 | 9.5 | feasible |
| D | 0 | 0 | 0 | 0 | 0 | infeasible |
| E | 0 | 0 | 0 | 0 | 0 | infeasible |
| F | 0 | 0 | 0 | 0 | 0 | infeasible |

According to the comprehensive feasibility evaluation results, schemes B and C were acceptable, while schemes D, E and F were not feasible. Therefore, the appropriate scale of reclamation in Haizhou Bay was 64 km$^2$ (Table 17).

**Table 17.** The maximum allowable area for CR in Haizhou Bay (scenario 2).

| Reclamation Scheme | Evaluation Result | Area (km$^2$) | Appropriate Scale of Reclamation (km$^2$) |
|:---:|:---:|:---:|:---:|
| A | - | 0.00 | |
| B | feasible | 18 | |
| C | feasible | 64 | |
| D | infeasible | 86 | 64 |
| E | infeasible | 105 | |
| F | infeasible | 114 | |

*3.3. Scenario 3 (S3)*

In scenario 3, HZ1 and HZ3 were superimposed first, then HZ5 and HZ2, superimposing from HZ1, HZ3, HZ5, HZ2 to HZ4 (Table 18).

**Table 18.** Changes in marine area of Haizhou Bay (scenario 3).

| Reclamation Scheme | Working Conditions | Reclamation Area/km$^2$ | Calculated Area of EC/km$^2$ |
|:---:|:---:|:---:|:---:|
| A | Current situation (no working condition) | 0.00 | 876.39 |
| B | HZ1 | 18 | 858.39 |
| C | HZ1+HZ3 | 64 | 812.39 |
| D | HZ1+HZ3+HZ5 | 83 | 793.39 |
| E | HZ1+HZ2+HZ3+HZ5 | 105 | 771.39 |
| F | HZ1+HZ2+HZ3+HZ4+HZ5 | 114 | 762.39 |

3.3.1. Calculation Area of EC

In scenario 3, the coastal shape was also superimposed according to working conditions to calculate the EC of pollutants in Haizhou Bay (Figure 9).

In scenario 3, after the implementation of various reclamation projects, the sea area of Haizhou Bay will change (Table 18).

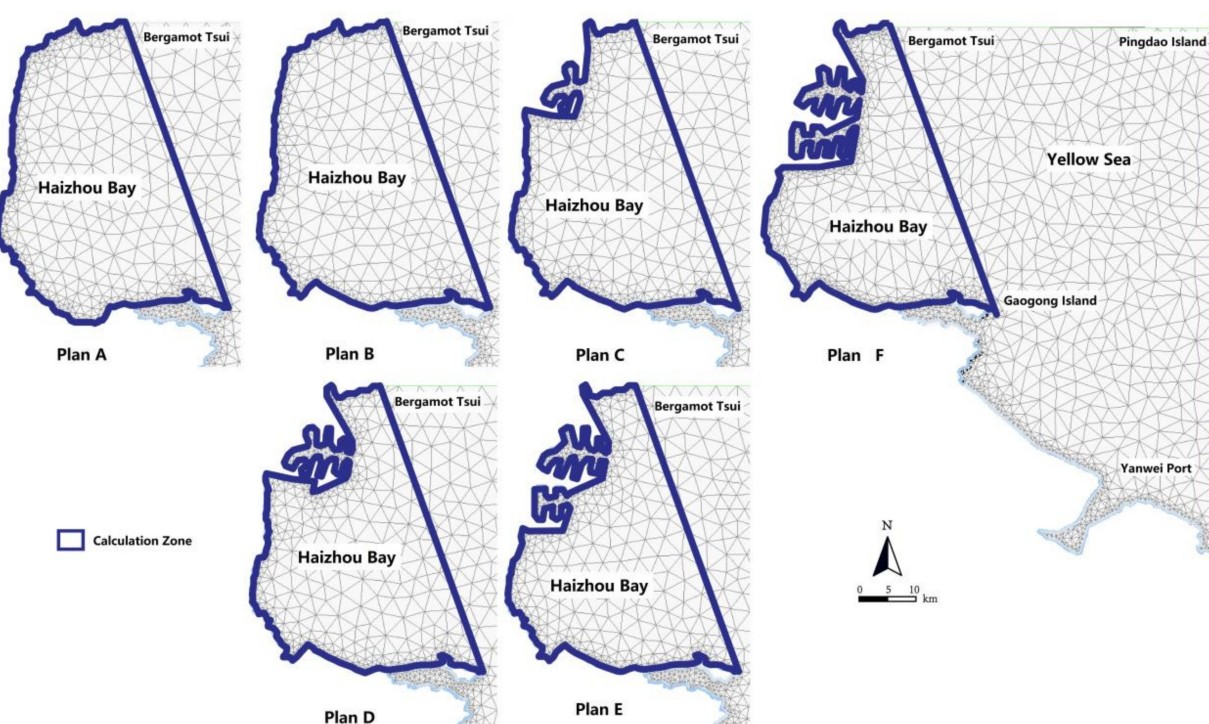

**Figure 9.** Computed region of marine EC (scenario 3). Computed region of plan A; computed region of plan B; computed region of plan C; computed region of plan D; computed region of plan E; computed region of plan F.

### 3.3.2. COD

Similar to scenario 2, after implementation of schemes B and C, the average tidal volume in Haizhou Bay decreased by 1.48% and 0.15%, respectively (Table 19). Different from scenario 1 and 2, after the implementation of scheme D, COD environmental capacity reduced by 8.67%. It appears that the position of working conditions is very crucial, which directly determines the diffusion of pollutants and the EC. After the sequential implementation of schemes E and F, the reduction ratio of COD environmental capacity reached 22.20% and 16.22%, respectively. These two schemes were the same as for scenario 2, so the change in EC was also consistent.

**Table 19.** Feasibility evaluation of COD environmental capacity.

| Reclamation Scheme | COD (t/a) | EC Change (%) | $R_1$ |
|:---:|:---:|:---:|:---:|
| A | 62,444 | - | - |
| B | 61,517 | 1.48 | 10 |
| C | 62,350 | 0.15 | 10 |
| D | 57,033 | 8.67 | 4 |
| E | 48,584 | 22.20 | 0 |
| F | 52,314 | 16.22 | 0 |

The feasibility evaluation results of EC demonstrated that the evaluation index $R_1$ of schemes B and C was 10, so scheme B and C were feasible. The evaluation index $R_1$ of scheme D was 4, and scheme D was also basically feasible. The evaluation index $R_1$ of schemes E and F was 0, and schemes E and F were not feasible, that is, working conditions HZ2 and HZ4 were prohibited.

### 3.3.3. $NH_4$-N

In scenario 3, the order of reclamation working conditions of schemes B and C was the same as for scenario 2, so the impact on EC of $NH_4$-N was the same as in scenario 2, being 4.45% and 0.62%, respectively (Table 20). In the implementation of scheme D, which was different from the first two scenarios, the impact on EC also changed due to different working conditions. Schemes E and F were the same as in scenario 2, but had a greater impact on the EC, with a reduction of more than 15%.

**Table 20.** Feasibility evaluation of $NH_4$-N environmental capacity.

| Reclamation Scheme | $NH_4$-N (t/a) | EC Change (%) | $R_2$ |
|---|---|---|---|
| A | 3366 | - | - |
| B | 3217 | 4.45 | 8 |
| C | 3346 | 0.62 | 10 |
| D | 3057 | 9.20 | 8 |
| E | 2673 | 20.59 | 0 |
| F | 2831 | 15.90 | 0 |

The feasibility evaluation results of $NH_4$-N environmental capacity demonstrated that the evaluation index $R_2$ of schemes B and D was 8, and scheme B and D were basically feasible. The evaluation index $R_2$ of scheme C was 10, so scheme C was feasible. The evaluation index $R_2$ of schemes E and F was 0, so scheme E and F were not feasible, and working conditions HZ2 and HZ4 were prohibited.

### 3.3.4. $NO_X$-N

With the same impact on the EC as $NH_4$-N, the order of reclamation working conditions of schemes B and C were the same as for scenario 2, so the impact on the EC of $NH_4$-N was consistent with scenario 2 (Table 21). Similarly, in the implementation of scheme D, the impact on $NO_X$-N environmental capacity was also reduced to 8.75%. In schemes E and F, as in scenario 2, the EC was greatly reduced, reaching more than 16%.

**Table 21.** Feasibility evaluation of $NO_X$-N environmental capacity.

| Reclamation Scheme | $NO_X$-N(t/a) | EC Change (%) | $R_3$ |
|---|---|---|---|
| A | 5082 | - | - |
| B | 4887 | 3.84 | 10 |
| C | 5045 | 0.07 | 10 |
| D | 4637 | 8.75 | 4 |
| E | 4250 | 16.37 | 0 |
| F | 4021 | 20.88 | 0 |

The feasibility evaluation results of $NO_X$-N environmental capacity indicated that the evaluation index $R_3$ of schemes B and C was 10, and schemes B and C were feasible. The evaluation index $R_3$ of scheme D was 4, so scheme D was basically feasible. The evaluation index $R_3$ of schemes E and F was 0, that is, working conditions HZ2 and HZ4 were prohibited.

### 3.3.5. $PO_4$-P

As for the first three pollutants, the EC of $PO_4$-P was slightly affected by scheme B (2.58%), moderately affected by scheme C (5.19%), slightly increased by scheme D (9.94%), and greatly affected by schemes E and F, with a reduction of more than 15%. The overall $PO_4$-P environmental capacity was affected by reclamation schemes with a gradually increasing trend (Table 22).

The EC feasibility evaluation results demonstrated that the evaluation index $R_4$ of scheme B was 10, and the evaluation index $R_4$ of schemes C and D was 4, and the three

schemes were feasible, but related environmental protection measures would be needed. Similar to the first three pollutants, schemes E and F were not feasible, and working conditions HZ2 and HZ4 were prohibited.

**Table 22.** Feasibility evaluation of $PO_4$-P environmental capacity.

| Reclamation Scheme | $PO_4$-P (t/a) | EC Change (%) | $R_4$ |
| :---: | :---: | :---: | :---: |
| A | 421 | - | - |
| B | 410 | 2.58 | 10 |
| C | 399 | 5.19 | 4 |
| D | 379 | 9.94 | 4 |
| E | 354 | 15.88 | 0 |
| F | 332 | 21.11 | 0 |

### 3.3.6. Comprehensive Evaluation of Feasibility

According to the analysis of the above four indicators, both schemes E and F had a greater impact on the EC of pollutants (Table 23), so the comprehensive evaluation index $R$ was 0. The influence of schemes B, C and D on the EC of the four pollutants was within the acceptable range, and the comprehensive evaluation index $R$ was 9.5, 8.5 and 5, respectively. Therefore, working conditions HZ1, HZ3 and HZ5 were feasible. However, HZ2 and HZ4 were both forbidden.

**Table 23.** Feasibility evaluation of EC (scenario 3).

| Reclamation Scheme | $R_1$ | $R_2$ | $R_3$ | $R_4$ | $R$ | Evaluation Result |
| :---: | :---: | :---: | :---: | :---: | :---: | :---: |
| A | - | - | - | - | - | - |
| B | 10 | 8 | 10 | 10 | 9.5 | feasible |
| C | 10 | 10 | 10 | 4 | 8.5 | feasible |
| D | 4 | 8 | 4 | 4 | 5 | basically feasible |
| E | 0 | 0 | 0 | 0 | 0 | infeasible |
| F | 0 | 0 | 0 | 0 | 0 | infeasible |

According to the comprehensive feasibility evaluation results, schemes B, C and D were acceptable, while schemes E and F were not feasible. Therefore, the appropriate scale of reclamation in Haizhou Bay was 83 km$^2$ (Table 24).

**Table 24.** The maximum allowable area for CR in Haizhou Bay (scenario 3).

| Reclamation Scheme | Evaluation Result | Area (km$^2$) | Appropriate Scale of Reclamation (km$^2$) |
| :---: | :---: | :---: | :---: |
| A | - | 0.00 | |
| B | feasible | 18 | |
| C | feasible | 64 | |
| D | basically feasible | 83 | 83 |
| E | infeasible | 105 | |
| F | infeasible | 114 | |

### 3.4. Comprehensive Analysis

Taking the COD response coefficient field formed by pollution point sources as an example (Figure 5), it was found that reclamation activities led to changes in the pollutant response field formed by each pollution point source. With the superposed implementation of reclamation activities, the response coefficient gradually increased. This also demonstrated that the pollution in Haizhou Bay was gradually increasing due to the overlapping implementation of the activities.

By analyzing the changes of pollutant EC caused by the implementation of each scenario, it was found that these reclamation projects had a spread impact on the EC of

the four pollutants, and the impact on $NH_4$-N and $PO_4$-P was greater than on COD and $NO_X$-N. In addition, the change of EC of pollutants had an extensive relationship with the location of the implementation of the reclamation schemes. For the same reclamation project, due to different scenarios, the sequence and location of the reclamation changed, leading to strong changes in EC.

According to the comprehensive evaluation and analysis of the three scenarios (Table 25), the conclusions were as follows. For scenario 3, schemes B, C and D were all permissible, and the suitable area for reclamation was 83 km$^2$, which was the scenario with the least impact on environment capacity, among all reclamation scenarios. Therefore, scenario 3 is recommended as the best reclamation scheme. In scenario 2, schemes B and C were allowed to be implemented, and the suitable area for reclamation was 64 km$^2$, which is recommended as the second-best reclamation scenario. In scenario 1, only scheme B could be implemented and the reclamation suitable area for reclamation was the smallest (18 km$^2$). Therefore, it was the worst reclamation scenario and is not recommended.

**Table 25.** Comprehensive evaluation of the three scenarios for CR.

| Reclamation Scheme | Scenario 1 $R$ | Scenario 2 $R$ | Scenario 3 $R$ |
|---|---|---|---|
| B | 9.5 | 9.5 | 9.5 |
| C | 0 | 9.5 | 8.5 |
| D | 0 | 0 | 5 |
| E | 0 | 0 | 0 |
| F | 0 | 0 | 0 |
| Appropriate area of reclamation (km$^2$) | 18 | 64 | 83 |
| Preferred scenario | 3 | 2 | 1 |

## 4. Conclusions

This paper analyzed the appropriate scale of reclamation in Haizhou Bay based on the impact of reclamation on EC. According to the water pollution status of Haizhou Bay and the location of reclamation conditions, the EC of four major pollutants (COD, $NH_4$-N, $PO_4$-P and $NO_X$-N) in Haizhou Bay were selected in this study to calculate the optimal scale of reclamation. The reclamation schemes had a large influence on the EC of the four pollutants, and the influence of $PO_4$-P and $NH_4$-N on the EC, was greater than COD and $NO_X$-N. From the perspective of the impact of alternative scenarios on the EC, scenario 3 had the least impact on the EC of pollutants, scenario 2 was second, and scenario 1 had the greatest impact on the EC. Therefore, scenario 3 was the first choice for reclamation, scenario 2 was the alternative, and scenario 1 was the worst option. From the perspective of the allowable reclamation area of the alternative scenarios, scenario 3 had the largest suitable reclamation scale of 83 km$^2$, scenario 2 was 64 km$^2$, ranking second, and scenario 1 had the smallest reclamation scale of 18 km$^2$. Overall, scenario 3 was the optimal reclamation scheme. Under the constraint of EC, the appropriate area of reclamation was found to be 83 km$^2$.

Since reclamation is a complex project system involving complex issues of multiple factors, there are many uncertainties in the process of pre-planning, mid-term construction and post-application. The suitable size estimated in this study was calculated based on the implementation of the actual reclamation project, therefore, there is still a certain amount of error. Any coastline is formed through the accumulation of long-term geological activities and development processes. A coastal reclamation project is an artificial change of a long-established geological landscape within a short period of time, which is necessarily a complex project involving various factors on land and sea. The issue of the appropriate scale of reclamation has not been systematically studied at home or abroad. Due to a lack of monitoring data, changes in the geomorphology of the seafloor caused by the reclamation project and the increase in other pollutants due to the reduction in environmental capacity caused by the reclamation, are not addressed in this study. How to explore a systematic

and comprehensive method for solving the appropriate scale of reclamation in the absence of detailed historical data, and under the existing socio-economic development rate, is the main thrust of establishing a comprehensive evaluation method for the environmental impact of reclamation projects in China.

**Supplementary Materials:** The following supporting information can be downloaded at: https://www.mdpi.com/article/10.3390/buildings12101673/s1. Figure S1: Verification point distribution in tidal flow field; Figure S2: Comparison of Simulated and measured tidal level in sea area; Figure S3: Relative error of simulated and measured tidal level; Figure S4: Tide Verification (Spring tide); Figure S5: Tide Verification (neap tide); Figure S6: Relative error of simulated and measured tide; Figure S7: Distribution of verification points on water quality; Figure S8: Comparison of simulated and measured data of COD concentration; Figure S9: Comparison of simulated and measured data of NH4-N concentration; Figure S10: Comparison of simulated and measured data of NOX-N concentration; Figure S11: Comparison of simulated and measured data of PO4-P concentration; Figure S12: Relative error of pollutant concentration simulated and measured values; Table S1: Pollutant loads from main sewage outfalls in Haizhou Bay (2020); Table S2: Pollutant loads from main marine estuaries in Haizhou Bay (2020).

**Author Contributions:** L.F.: conceptualization, methodology, software; X.Z.: data collection and curation, writing—original draft, preparation; P.H.: data collection and curation, software, validation; X.X.: data collection and curation, writing—original draft. All authors have read and agreed to the published version of the manuscript.

**Funding:** This work was supported by the Overseas Study Fellowship from the China Scholarship Council, National Natural Science Foundation of China (No. 42007151), the Philosophy and Social Science Foundation of Colleges and Universities of Jiangsu Province (No. 2019SJA0108), the Natural Science Foundation of Colleges and Universities of Jiangsu Province (No. 19KJB610015), and the Social Science Application Research Project of Jiangsu Province (No. 19SYC-067). The authors wish to thank Nanjing University for providing MIKE software for use in this study.

**Institutional Review Board Statement:** Not applicable.

**Informed Consent Statement:** Not applicable.

**Data Availability Statement:** Not applicable.

**Conflicts of Interest:** The authors declare no conflict of interest.

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
