# Peer review of "Exploring the Optimal Scale of Coastal Reclamation Activities Based on an Environmental Capacity Assessment System: A Case Study in Haizhou Bay, China"

_buildings, doi:10.3390/buildings12101673_

Round 1

Reviewer 1 Report

The topic is important, not new, however the question is: is there enough data and is it reliable for the application of the method? because if not, it is just an academic exercise with no real application. If there is insufficient data, the title of the article is not met, hence it should be reformulated from the title of the article since the expectations generated by it are not met. The wording tends to be repetitive as lines 51 to 58 talk about damage and changes of use. It is repeated from 65 to 82. I suggest joining the two paragraphs so that it does not become repetitive. Line 418 talks about data that is not in the article There is an estimate of the size of the error that the study could have as mentioned in line 766

Reviewer 2 Report

Costal reclamation is an effective way to tackle the contradiction between the land shortage and the living space demand for human beings. Its benefits and problems are increasingly obvious in the 21st century. To harmonize the contradiction, it is necessary to develop a suitable scheme for costal reclamation. Based on the natural conditions and social and economic conditions of Haizhou Bay, the author established ECAS assessment method. Four major pollutants were used to calculate the optimum reclamation area for each specific sea zone and the optimal reclamation scenario was confirmed. This study could be a useful reference to management of coastal reclamation and sustainable land use in coastal areas. The article structure is well organized. The conclusions reported in the article are also objective and accurate. English language style is also very used. Based on the current status, I personally think, a minor revision is required before possible publication in the journal buildings. Here are some detailed suggestions for further modification:

Major comments:

1.     Figure 1, the RGB band and time used for the image should be added. The bounder legend should also be added.

2.     In “2.2.4. Comprehensive assessment”, the definition of “Evaluation score” (Table 3) is not described in the content. And why can R be used as a criterion of feasibility? Why did four pollutants get the same weight in the calculation of R, any previous ref?

3.     The content about MIKE 21 is somewhat redundant, the part of parameter setting can be simplified.

4.     In “3 Result and Discussion”, the Reclamation plans (A, B, F) in three scenarios are similar. The distribution about four pollutants is described in detailed, with too many tables used here.

5.     The definition of Scenario 1-3 is not mentioned in the previous sections.

Minor comments:

1.     Line50, the cases of “anthropogenic activities” don’t seem quite appropriate under this backgroud.

2.     Line86, change “eenvironment” with “environment”.

3.     Line436-439, the font size is different.

4. I didn't see the uploaded supplementary materials.
